# High Temperature Magnetic Cores Based on PowderMEMS Technique for Integrated Inductors with Active Cooling

**DOI:** 10.3390/mi13030347

**Published:** 2022-02-22

**Authors:** Malte Paesler, Thomas Lisec, Holger Kapels

**Affiliations:** Fraunhofer Institute for Silicon Technology, Fraunhoferstrasse 1, 25524 Itzehoe, Germany; thomas.lisec@isit.fraunhofer.de (T.L.); holger.kapels@isit.fraunhofer.de (H.K.)

**Keywords:** integrated inductances, PowderMEMS technique, magnetic core, atomic layer deposition, high temperature, integrated DC/DC converter, porous structure, active cooling

## Abstract

The paper presents the realization and characterization of micro-inductors with core with active cooling capability for future integrated DC/DC converter solutions operating with wide bandgap semiconductors at high temperatures with high power densities. The cores are fabricated backend-of-line compatible by filling cavities in silicon wafers with soft magnetic iron particles and their subsequent agglomeration to rigid, porous 3D microstructures by atomic layer deposition. Wafer processing is presented as well as measurement results at up to 400 ${}^{\circ }C$ operating temperature in comparison to of-the-shelf inductors. Using a DC/DC converter operating at 25 MHz switching frequency efficiencies of 81 to 83% are demonstrated for input voltages between 5 V and 12 V. It is shown that the temperature of the novel micro-inductors decreases if an air flow through its porous core is applied. This feature could be especially helpful for the realization of resonant power converters with larger temperature stress to passive components.

## 1. Introduction

The miniaturization of passive components is a key factor to reach smaller package sizes and higher power densities for DC/DC converters. The idea of a power supply on a chip (PwrSoC) [1] or a power supply in a package (PwrSiP) [2] strives for innovative integration solutions. Especially with the new generation of wide band-gap devices like silicon carbide and gallium nitride transistors, higher switching frequencies are possible and fewer switching losses occur at the same time. Switching frequencies in the range of 10–50 MHz allow inductances below 200 nH in typical DC/DC converters [3,4]. Thus, MEMS fabrication processes become a real option to establish inductances on silicon and enable valuable integration solutions with transistors, diodes, etc. on the same substrate, coming along with the benefits of low parasitics and proper thermal management.

In [1,5,6,7], several processes and designs of inductors on silicon wafers were discussed, but mainly the windings were placed on the surface like spiral inductors. Towards further device integration, 3D-inductors were developed with an air core and through-silicon vias (TSVs) in [8,9]. These 3D inductors have already improved inductance density but still at low levels. Ref. [10] describes a novel type of magnetic cores for integrated inductances, which can be fabricated in a back-end-of-line (BEOL) compatible manner using PowderMEMS. This novel microfabrication technique is based on the solidification of micron-sized soft magnetic powder by means of atomic layer deposition (ALD) [11]. In [12] the impact of different soft magnetic powders on the performance of inductors with powder-based cores and manually wound wires was evaluated. A boost converter with a GaN FET was developed to prove the functionality of selected cores. At 20 MHz an input voltage of 15 V could be boosted to 25 V on the output at a load current of 481 mA with an efficiency of 87%.

In this paper the magnetic characteristics of selected powder-based cores are investigated within a temperature range up to 400 ${}^{\circ }C$ and compared to the performance of commercially available reference inductors. The possibility of active cooling an inductor with powder-based core by an air flow through it is demonstrated. As target application, an improved DC/DC converter with the micro-inductor is set up. At a switching frequency of 25 MHz, this prototype converts 10 W with an efficiency of 83%.

## 2. Fabrication Process

The cross-sections in Figure 1 schematically illustrate the fabrication of silicon chips with integrated core and fluidic access openings on standard 8-inch silicon substrates.

After oxidation of the silicon substrate (650 nm SiO${}_{2}$) and the 1st lithography, about 500 μm deep cavities, defining the core dimensions, are created by Deep Reactive Ion Etching (DRIE) (Figure 1a). Without removal of the photo resist mask used for DRIE, the substrates are now transferred from the MEMS cleanroom into a dedicated lab, where the porous soft magnetic cores are fabricated using the PowderMEMS technique [13]. Here, in a first step soft magnetic powder (Carbonyl Iron Powder YTF-HY3 from PMCTec, d50 = 4.3 μm) is dry filled into the cavities (Figure 1b). In a second step, the loose dry particles are agglomerated into rigid 3D micro-structures with 75 nm Al${}_{2}$O${}_{3}$, deposited by ALD at 75 ${}^{\circ }C$. The ALD layer coats all particles throughout the particle bed as well as walls and bottom of the cavity. Within the cross-section in Figure 1c, this covering is highlighted in red. Thanks to that, neighboring particles are connected to each other at the points of contact as well as to the cavity surfaces, forming a rigid, porous 3D microstructure over the entire cavity volume. The substrates with the embedded soft magnetic cores are now subjected to a surface conditioning procedure to regain cleanroom compatibility. Firstly, grinding and polishing are applied to remove stuck particles from the backside of the substrate. Secondly, after transfer back into the cleanroom, the remaining photoresist on the front side is removed together with stuck particles by treatment in O${}_{2}$ plasma, subsequent lift-off in organic solvent and final cleaning. To seal the porous soft magnetic cores a 3 μm thick silicon oxide is deposited by plasma enhanced chemical vapor deposition (PECVD) at 300 ${}^{\circ }C$ (Figure 1d). After the 2nd lithography the PECVD layer is patterned by RIE to open the areas where through-openings along the core must be created within the Si substrate to facilitate wire wounding (not illustrated in the cross sections). Then the resist mask is removed and the substate is turned upside down. After the 3rd lithography, access holes are etched into the porous soft magnetic core by DRIE. During the same process the through openings along the core are created. Within the fluid access holes the DRIE process securely stops at the Al${}_{2}$O${}_{3}$ ALD layer on the former bottom of the cavity thanks to its outstanding etching selectivity (Figure 1e). Finally, the Al${}_{2}$O${}_{3}$ is removed by RIE to enable a fluid flow through the porous soft magnetic core, see Figure 1f.

Figure 2 depicts SEM images of top and bottom side of a finished silicon chip after dicing of the wafer. The openings on the backside of the magnetic core are dimensioned in such a way that fluid connectors can be attached (Figure 2 (right)). Figure 3 shows a SEM micrograph of a cross-sectional polish through such a chip in longitudinal direction.

Selected chips were then picked and 100 μm thick wire was wounded manually to complete the micro-inductor. For cooling tests some of those micro-inductors were glued topside-down to a glass plate for better fluidic sealing. After that, glass tubes were mounted above the openings on the bottom side to enable a fluid flow through the porous core. The drawing in Figure 4 illustrates the flow-through setup schematically.

## 3. Simulation Model

Finite element method (FEM) simulations were used to support the design and optimization process of these micro-inductors. In this case, Comsol multiphysics is the software to model electro-magnetic physics based on Maxwell’s equations and the heat transfer interface to investigate thermal effects caused by the current through the windings. First, the geometry of the fabricated test sample was created (Figure 5). To determine the exact size and respectively the volume of the magnet core, scanning electronic microscopy was used. Due to the fact of a hand-wound inductor, the distances between the windings were measured and averaged. In the simulation model, the test sample was surrounded by a block of air, which size was defined as a comprise between precise results and fast computation time with focus on the accuracy.

### 3.1. Electromagnetic Study

Mesh creation is an important aspect for producing reliable results. Especially when analysis are done for a wide frequency range, parameters like e.g., the skin depth are changing dramatically. Because of frequency studies in the range between 100 kHz to 100 MHz, the focus during mesh creation was on skin and proximity effect related areas like the boundaries of the windings.

In [10], a new parameter set for the powder-based core was developed, a software library created and verified with permeability and electrical parameter measurements. The new software library was used to develop these new inductor samples with additional cooling interfaces. A pre-study of the effectiveness of active cooling as well as the determination of the amount of windings to reach the proper inductance for the desired application was performed by simulation. Simulated and experimentally obtained inductance and resistivity parameters matched well. The corresponding measurement results for the switching frequency of 25 MHz of the target application are presented in Section 6.1.2.

### 3.2. Thermal Study

In addition to the electromagnetic investigations, the described simulation model was extended by a thermal interface to determine the component temperatures as a function of the applied load and cooling conditions. In contrast to typical frequency studies with a sinusoidal waveform, this simulation was run in the time domain with application-like triangular current characteristics through the windings. For the main investigations of the behavior of the prototype, the coil current was varied between 0.24 A and 1.36 A with a frequency of 25 MHz (see also the next chapter for further details). As the high frequency current variation requires simulation steps in the nano-second range and, on the other hand, the much smaller rate of change of the temperature requires an application simulation time of ten to hundreds of seconds unless a stable operating point is reached, a combined simulation approach was used in Comsol. The power loss densities were calculated in a frequency study, followed by a stationary study for the thermal simulation. For the thermal simulations the model as shown in Figure 5 has been extended by a glass plate underneath the inductor and two glass tubes above the fluidic openings, according to Figure 4. It is assumed that this setup is surrounded by air. Only the ends of the wire, which are soldered to the PCB on the prototype board, were fixed to a temperature of 40 ${}^{\circ }C$ and 50 ${}^{\circ }C$, as they were measured in the DC/DC converter setup. In Figure 6 (left), the surface temperature in steady state is displayed and a maximum temperature of 81.5 ${}^{\circ }C$ is reached.

#### Active Cooling

To simulate active cooling by a fluid flow through the porous magnetic core, the model was further expanded by the so-called Free and Porous Media Flow interface. The first investigations were performed for air as coolant. As shown in Figure 6 (right), the air is pumped with a velocity of 1 cm/ s into the tube on the left side and leaves through the tube on the right side after passing the porous core in the middle. Based on the investigations in [14], the non-darcy permeability model was chosen. The porosity of the core was set to 0.5 in accordance with our SEM analysis. Another important parameter of the core material is the intrinsic permeability *k* in $m^{2}$, defined as
(1)$$k=K\frac{\eta }{\rho g},$$
where *K* is the hydraulic conductivity in m/s, η is the dynamic viscosity of the fluid in Pa·s, ρ is the density of the fluid in kg/ $m^{3}$ and *g* is the acceleration due to the earth gravity in m/ $s^{2}$. After first experiments, the permeability was estimated to be approximately $k=1\times 10^{-11}\;m^{2}$. In comparison to the simulation without active cooling, this simulation results in a maximum temperature of 74.1 ${}^{\circ }C$, which means a drop of 7.4 ${}^{\circ }C$ due to the active air cooling.

## 4. Characterization

This section describes the utilized test setups. Three types of ISIT samples were investigated. For VSM measurements silicon chips with powder-based core without flow-through option as described in [10] were utilized. Impedance measurements were performed on micro-inductors, as shown in Figure 5, based on chips with flow-through core, as shown in Figure 2. Apart from core test samples and micro-inductors, for cooling tests flow-through setups were built from selected micro-inductors. For comparison, three commercially available inductors were bought and characterized in the same manner as the ISIT samples. All inductors should have a target inductance of 150 nH, further parameters are listed in Table 1. Pictures were taken by a 3D X-ray tomography.

### 4.1. Vibrating Sample Magnetometer Test Setup

The LakeShore VSM 7400 system (Figure 7 (left)) was used to record the magnetic moment *m* of a sample depending on the applied magnetic field *H*. The sample was mounted on the end of a rod, which hangs between the big pole shoes of the VSM. After some rotation calibration, the long side of the sample was orientated parallel to the direction of the magnetic field from the pole shoes. During the measurement, the rod is moved up and down with a frequency of 80 Hz. Therefore, the magnetic moment could be measured with four pick-up coils, which are located at the ends of the pole shoes. Before start of the measurement, a calibration is required to ensure that the sample is in the middle between these four pick-up coils. Basically, saturation flux density, remanent magnetization, magnetic coercivity and permeability are the main parameters to investigate.

#### Temperature Stable Core

One of the benefits of the PowderMEMS is, that in contrast to other powder-based techniques, no organic materials are needed for microstructure fabrication. Since the fabricated powder-based cores do not contain any organics, they exhibit increased thermal stability. That is of particular interest for applications, where the components have to withstand very high temperatures, e.g., LED drivers. To investigate the performance of the samples at elevated temperatures, measurements were executed on the VSM at 25 ${}^{\circ }C$, 100 ${}^{\circ }C$, 150 ${}^{\circ }C$, 300 ${}^{\circ }C$ and 400 ${}^{\circ }C$. The test run with a duration of 4 h started at 25 ${}^{\circ }C$ and was finished at 400 ${}^{\circ }C$. As shown in Figure 7 (right), for that an oven setup, operating under vacuum, is mounted in the VSM. After removal of the windinds two commercially available inductors were measured and compared to the ISIT core test samples based on the Carbonyl Iron Powder YTF-HY3 from PMCTec (d50 = 4.3 μm). Before the VSM measurements were started, the components were pretested on a heat plate to observe any optical degradation and to ensure that no outgassing appears, which would pollute the inside of the vacuum tube. The ISIT sample did not show any optical changes, whereas the reference inductors showed some smoke coming from the area of the shield and insulation. Therefore, these parts were cleaned using fuming nitric acid. For the VSM measurements the samples were fixed with a thermal resistant epoxy on the rod.

### 4.2. Impedance Analyzer Test Setup

For the characterization with the impedance analyzer, 8 turns of copper wire were hand wound around the fabricated powder-based core (Figure 5) and fixed with some glue. The device characterization was performed on the Agilent 4294A impedance analyzer within the frequency range from 100 kHz to 100 MHz. As displayed in Figure 8, the test sample was connected as close as possible to the terminals to minimize parasitic effects. To connect the samples to the impedance analyzer, the insulation was removed and the ends of the wires were soldered with a tin/silver solder.

## 5. DC/DC Converter with Micro-Inductor

One main target application for PowderMEMS based micro-inductors are compact DC/DC converters. For the proof-of-concept, a boost converter with operating voltages below 50 V and output power up to 20 W was chosen (Figure 9).

The operating point was determined in the way that the converter runs in continuous conduction mode (CCM). As switch the EPC8009 Enhancement Mode Power Transistor fits quite well to this application and in combination with Texas Instruments’ LMG1020 driver switching frequencies up to 50 MHz can be reached. The EPC8009 is rated up to 65 V and a continuous current of 4 A. The micro-inductor was assembled on DC/DC converter board in the flow-through setup (Figure 4). Due to the limited thermal path of the micro-inductor soldered on the PCB, a moderate current was set up. Following equations were used for the design and main parameters are summed up in Table 2. In a boost converter, the step-up ratio between input and output voltage is given by the duty cycle *D* as follows
(2)$$\frac{U_{out}}{U_{in}}=\frac{1}{1-D}.$$

Rearranging (Equation 2) gives
(3)$$U_{out}=U_{in}\;\cdot \;\frac{1}{1-D}.$$

As load, an ohmic resistance (R=50 Ω) is connected to the output terminals. The current ripple $\Delta i_{L}$ of a boost converter in CCM is given by
(4)$$\Delta i_{L}=\frac{U_{in}D}{f_{sw}L}.$$

The green DC/DC converter PCB sits piggybag on the red LaunchPadXL from TI (Figure 10 (left)). For the first tests, the LaunchPad is used to generate the PWM signal for the GaN driver on the converter board. At this desired operating point, the PWM signal has a period of 40 ns with a high-level voltage (3.3 V) for 20 ns. So far, the system is run at open loop control and furthermore voltage feedback to the microcontroller is installed and could be used in future tests. The measurements were taken by the HDO4104 oscilloscope from LeCroy (2.5 Gs/s), using passive probes connected via coax cables to U.FL. sockets on the PCB. The inductor current is measured with a shunt resistor, while the input and output current is monitored by a current clamp (CP031 from LeCroy up to 100 MHz).

### Active Cooling

Due to the porous structure of the magnetic core, a breakthrough in active cooling of passive device structures with a gas or liquid fluid flow through the core is possible. A further benefit is, that also additional components like e.g., the GaN FET can be included. In Figure 10 (right), the micro-inductor in the flow-through setup is shown. During operation of the DC/DC converter, the active cooling of the micro-inductor can be switched on. For the first experiments, an external peristaltic pump was set up to generate an air flow with a velocity of 1 cm/ s through the tube, which has an inner diameter of 1 mm. The influence and effect of this active cooling method was measured with an infrared camera (FLIR A655sc) plotting the maximum inductor temperature over the time during converter operation.

## 6. Measurement Results

This section provides measurement results for various ISIT samples as well as three commercially available products (Table 1). First, the core properties are compared. Secondly, the inductors are characterized, followed by investigations of the DC/DC converter in operation, where the micro-inductor from ISIT is integrated as a component.

### 6.1. Core and Inductor

#### 6.1.1. VSM Temperature Measurements

The VSM measurements started with at the positive maximum of the magnetic field $H_{max}$ in saturation and decreasing afterwards the magnetic field strength stepwise until $-H_{max}$ is reached. From this point, the magnetic field *H* was increased again up to $H_{max}$, closing the full loop. For all three investigated samples different maximum magnetic fields $H_{max}$ were identified, as shown in Table 3. Additionally, the magnetic saturation flux density $B_{sat}$ and the maximum permeability $\mu_{max}$ were measured. During the measurement, the magnetic moment *m* of the test sample for each magnetic field *H* is measured, too. Dividing the magnetic moment by the volume of the magnetic core $V_{C}$, the magnetization *M* is calculated ($M=\frac{m}{V_{C}}$). Based on Maxwell’s equations, the magnetic flux density *B* is given as $B=\mu_{0}\left(H+M\right)$, where $\mu_{0}$ is the vacuum permeability ($4\pi \;\cdot \;10^{-7}\frac{Vs}{Am}$).

In Figure 11a the measurement results are shown for a temperature of 25 ${}^{\circ }C$. The ISIT sample (red curve) has the highest magnetic saturation flux density of $B_{sat}=2.02\;T$, while the reference inductor 1 (blue) and the reference inductor 2 (green) are already saturating at $B_{sat}=0.65\;T$ and $B_{sat}=0.42\;T$, respectively. In respect of permeability, the maximum values are reached at a magnetic field strength of $H=1-3\times 10^{4}\;A/m$ with relative permeabilities in the range between 6 and 8.5.

The investigations on degradation effects to the magnetic characteristics of each sample at higher temperatures is shown in the Figure 11b (ISIT sample), Figure 11c (reference 1) and Figure 11d (reference 2). The temperature behavior of the magnetic moment up to very high temperatures was investigated for the same variation of magnetic field strength at temperatures of 25 ${}^{\circ }C$, 100 ${}^{\circ }C$, 150 ${}^{\circ }C$, 300 ${}^{\circ }C$ and 400 ${}^{\circ }C$. The ISIT sample results, provided in Figure 11b, shows nearly no degradation across the whole temperature range, whereas reference inductor 1 starts to degrade noticable above 150 ${}^{\circ }C$ and reference inductor 2 shows a massive breakdown above 100 ${}^{\circ }C$. In Figure 12, the ratio of the magnetic saturation flux density relative to its value at 25 ${}^{\circ }C$ is compared for all samples. At 400 ${}^{\circ }C$, the ISIT sample degrades by only 5%.

#### 6.1.2. Impedance Analyzer Inductor

The frequency characteristic of the micro-inductors is one of the most important parameter. Therefore, three reference inductors and the micro-inductor from ISIT were compared from 100 kHz to 100 MHz on an impedance analyzer. The results were plotted for the inductance (Figure 13a), the resistance (Figure 13b) and the quality factor Q=2πf·L/R (Figure 13c) as a function of the frequency. Regarding the inductance, the ISIT sample shows a better stability over the whole frequency range compared to reference inductor 2 and 3, while reference inductor 1 is similar. Focusing on the resistance, reference inductor 2 has the fastest increase at higher frequencies, whereas reference inductor 3 with the air core shows the flatest increase. The ISIT sample and reference inductor 1 are in the mid-range, whereas the reference inductor 1 shows the highest DC resistance. As a result, the quality factor for reference inductor 1 is 37 at 25 MHz, followed by ISIT sample (29), reference inductor 3 (25) and reference inductor 2 (15).

### 6.2. DC/DC Converter with Micro-Inductor

The investigation of the micro-inductors in the DC/DC converter was performed by increasing the input voltage step-wise while the duty cycle and output load were kept on the same level. Besides voltage and current measurements by oscilloscope, the test board was monitored by the infrared camera. In Table 4 all measurements are summed up. At a switching frequency of 25 MHz, the efficiency of this boost converter was in the range of 81.0–82.7%.

In Figure 14, the results are plotted for the operating point of $U_{in}=12\;V$. The gate voltage $U_{gs}$ (black) is alternating between 0 V and 5 V with a period of T=40 ns. Due to fast switching and parasitic inductances of the circuit layout, the drain-source voltage (red) shows some ringing effects. Across the capacitor, the output voltage (blue) is quite stable at $U_{out}=22.8\;V$. In the bottom plot of Figure 14a, the measured currents are displayed. While the input and output currents, measured by a current clamp, are quite stable, the curve of the inductor current (red) is oscillating significant due to the direct shunt resistor measurement. Despite this, the triangular inductor current waveform varies between 0.18 A and 1.88 A, resulting in a current delta of $\Delta i_{L}=1.7\;A$, which is close to the theoretically calculated value based on Equation (Equation 4) of:(5)$$\Delta i_{L}=\frac{12\;V\;\cdot \;0.514}{25\;M\mathrm{Hz}\;\cdot \;154\;nH}=1.65\;A$$

The infrared camera measurement results in steady state are shown in Figure 14b with one marker at the inductor and another one at the GaN FET. The inductor reaches a maximum temperature of $T_{L}=109\;{}^{\circ }C$, while the maximum temperature of the GaN FET is even larger at $T_{T}=122\;{}^{\circ }C$.

#### DC/DC Converter with Active Air-Cooled Micro-Inductor

Before adding the air-cooled inductor sample to the DC/DC converter, some pretests on the basic micro-inductor sample regarding the air flow through the tubes, inlets, outlets and the porous structure were performed. The pump forced an air flow through the tube with a velocity of approximately 1 cm/s into the porous core material inside the silicon. At the outlet of the inductor, a second tube was connected the micro-inductor, the other end was put into a glass of water, where air bubbles were rising up. The connection between the glass tube and silicon opening is especially challenging and needs to be handled carefully. After the successful pretest, the inductor was installed within the DC/DC converter board. The converter was operating with $U_{in}=9\;V$ for some minutes until the temperature of the inductor stabilized at $81.5\;{}^{\circ }C$. In Figure 15, the change of the maximum measured temperature can be observed, whereas the active cooling was switched on at $t_{1}=130\;s$. Approximately 70 s later, a new steady state with $T=80.0\;{}^{\circ }C$ was reached and recorded by the thermal image in Figure 15 (right). Above the inductor, the tubes and pumped air at room temperature can be seen.

## 7. Discussion

The powder-based cores from ISIT, fabricated using PowderMEMS technique, could be of considerable interest for integrated solutions, especially in converter applications with high temperatures and power densities. First, the optical investigation of the SEM proofs the idea of a solid porous structure as a magnetic core based on PowderMEMS technique. During VSM measurement at temperatures up to 400 ${}^{\circ }C$, the agglomerated core did not show any defects and the magnetic characteristics show only some variation of 5% within 25 ${}^{\circ }C$ and 400 ${}^{\circ }C$. In contrast, the commercially available products were found to be limited in the operation temperature of 125 ${}^{\circ }C$ due to significant degradation (8% and 40% degradation). In addition, the charaterization on the impedance analyzer indicates a good performance of the ISIT sample, where only reference inductor 1 shows a higher quality factor. Comparing the volume of the magnetic core of those two samples, the ISIT sample offers a volume reduction by more than 50%.

The developed simulation models allow a good estimation of the micro-inductor properties and speed up the development process for further samples. Especially, the electro-magnetic simulation model and its material parameter are matching quite accurate with the measurement results. According to the thermal study, the results are looking promising but a few results need to be investigated further. In particular for the active cooling, further parameter and flow models should be compared and investigated for this target application.

For the parameter range of the manufactured micro-inductors, a boost converter operating at 25 MHz is a typical target application. Compared to other converters in these power and frequency range, the achieved efficiency above 80% is quite remarkable. The presented results with and without active cooling provide the fundamental proof of concept. Active cooling of micro-inductors by an air flow through the powder-based core has been successfully demonstrated. The difference between measured and simulated effect can be related to undefined leakages at the fluidic connections to the silicon chip. For future investigations with active cooling, the connection between tube and silicon chip has to be improved. Active cooling with a liquid through the porous core structure will be tested, which will result in much better cooling performance than air. For a full proof of the presented concept, micro-inductors with integrated wiring based on vertical feedthroughs will be developed.

## Figures and Tables

**Figure 1 micromachines-13-00347-f001:**
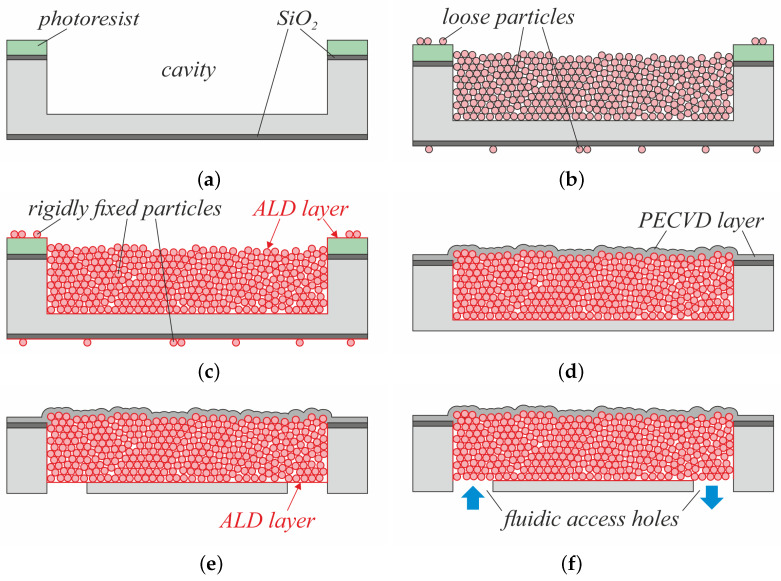
Schematic illustrations (**a**–**f**) of the fabrication process of test structures with soft magnetic flow-through cores on Si substrates.

**Figure 2 micromachines-13-00347-f002:**
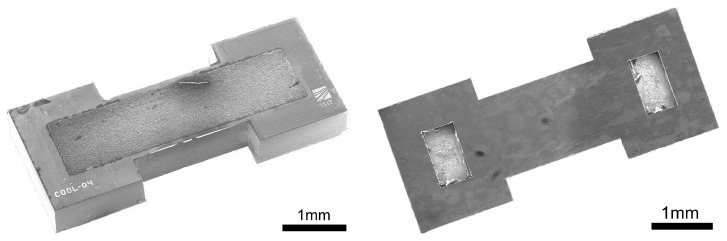
SEM micrographs of the top side (**left**) and the bottom side (**right**) of a finished silicon chip with embedded core and fluidic openings.

**Figure 3 micromachines-13-00347-f003:**
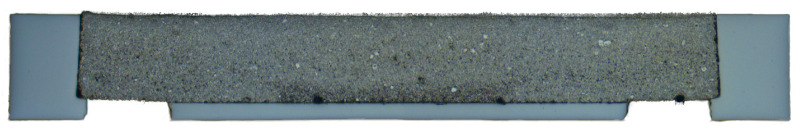
SEM micrograph of a cross-sectional polish through a silicon chip as shown in Figure 2.

**Figure 4 micromachines-13-00347-f004:**
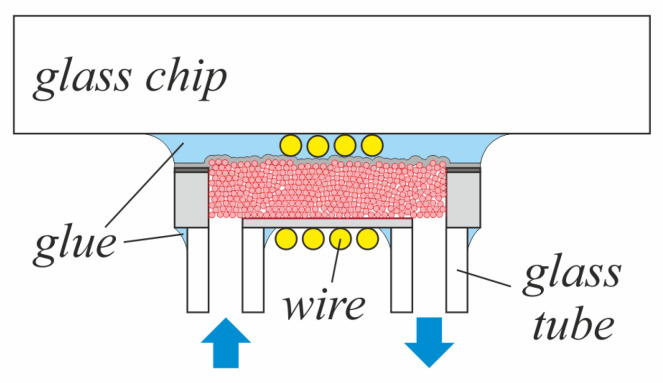
Flow-through setup: Cross-section (schematically) with additional glass plate and tubes for cooling tests.

**Figure 5 micromachines-13-00347-f005:**
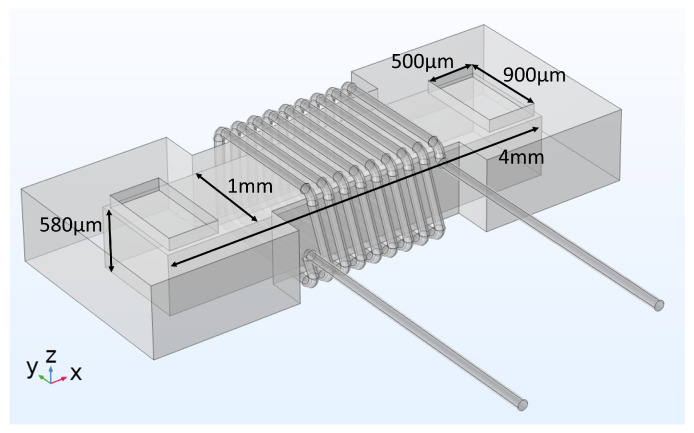
Model of the micro-inductor in Comsol Multiphysics.

**Figure 6 micromachines-13-00347-f006:**
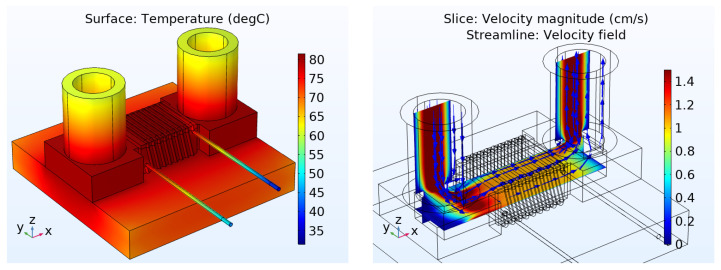
Simulation results: Surface temperature without cooling (**left**) and fluid velocity with active cooling (**right**).

**Figure 7 micromachines-13-00347-f007:**
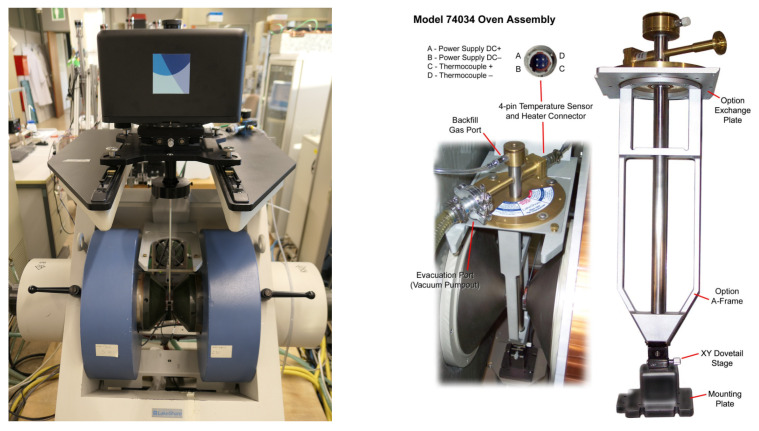
LakeShore VSM 7400 system (**left**) and Oven Assembly 74034 (**right**) [15].

**Figure 8 micromachines-13-00347-f008:**
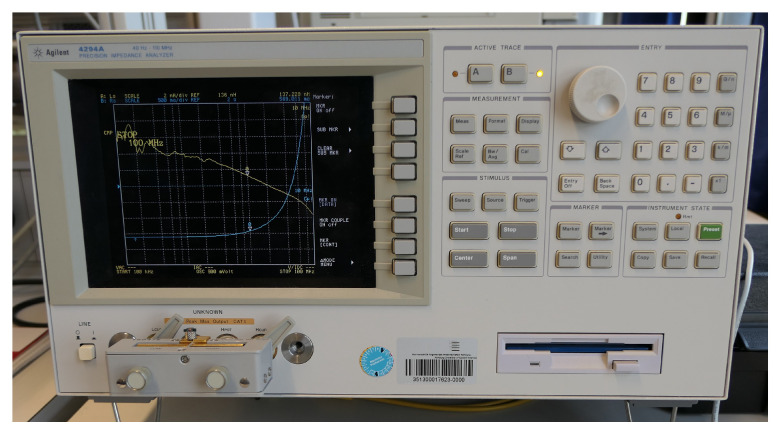
Impedance analyzer Agilent 4294A with the attached ISIT sample.

**Figure 9 micromachines-13-00347-f009:**
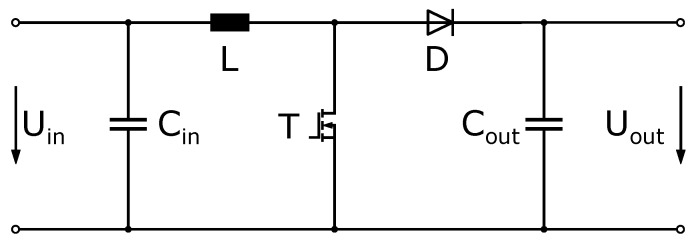
Topology of the prototype.

**Figure 10 micromachines-13-00347-f010:**
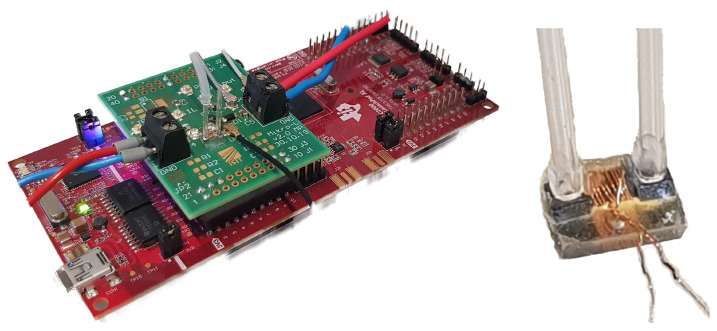
Picture of the DC/DC converter (**left**) assembled with micro-inductor (zoomed: **right**).

**Figure 11 micromachines-13-00347-f011:**
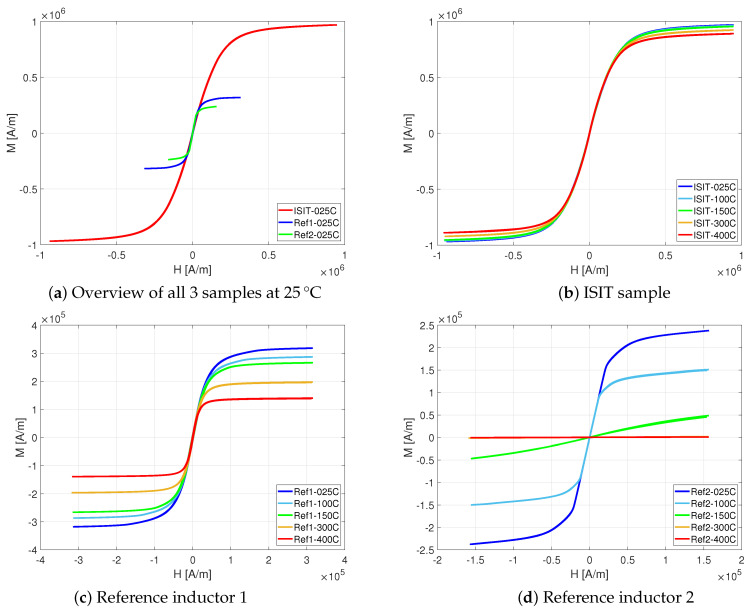
VSM temperature measurements.

**Figure 12 micromachines-13-00347-f012:**
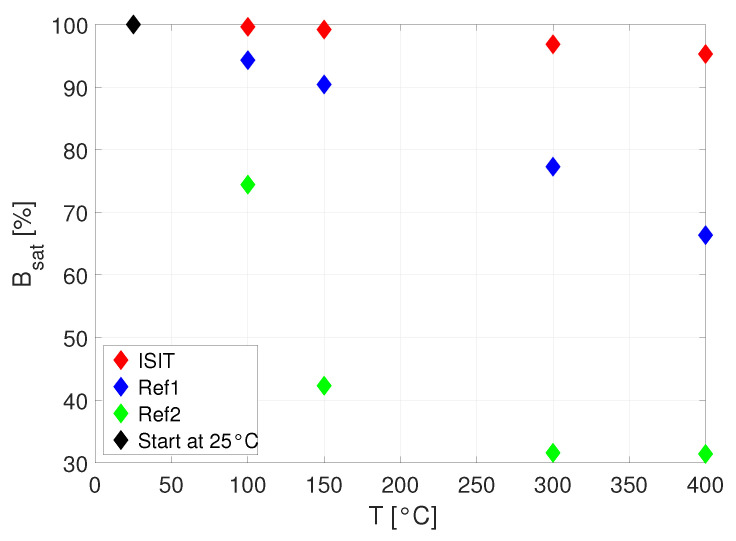
Magnetic saturation flux density $B_{sat}$ versus temperature.

**Figure 13 micromachines-13-00347-f013:**
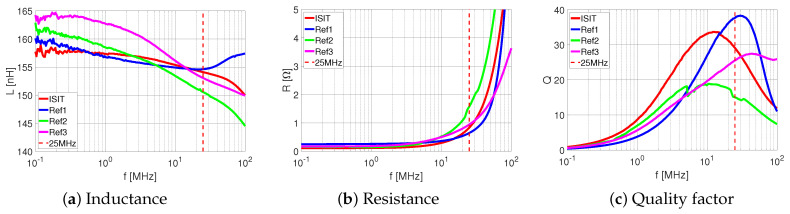
Measurement results from impedance analyzer.

**Figure 14 micromachines-13-00347-f014:**
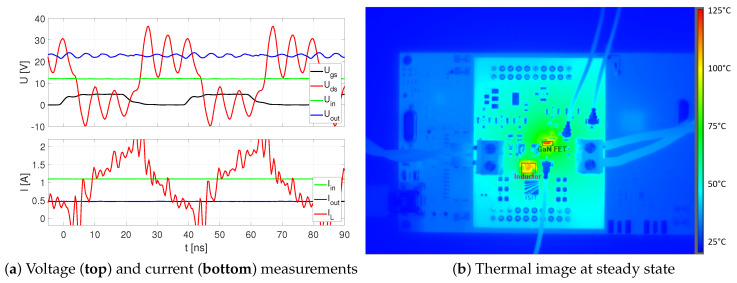
Measurement results of DC/DC converter at $U_{in}=12\;V$.

**Figure 15 micromachines-13-00347-f015:**
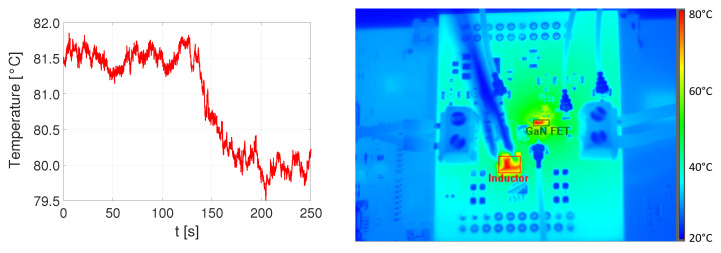
Temperature curve (**left**) and thermal image during active cooling (**right**).

**Table 1 micromachines-13-00347-t001:** Reference inductors.

	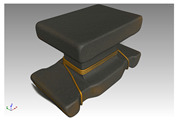	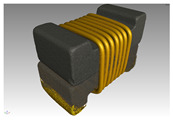	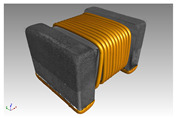
	Inductor Ref1	Inductor Ref2	Inductor Ref3
Core	ferrite	ceramic/ferrite	ceramic
DCR	390 mΩ	150 mΩ	220 mΩ
Imax	0.25 A	1.3 A	0.8 A
Tmax	85 ${}^{\circ }C$	85 ${}^{\circ }C$	125 ${}^{\circ }C$
Size [mm]	3.2 × 1.6 × 1.8	1.8 × 1.17 × 1.12	2.6 × 2.1 × 1.9

**Table 2 micromachines-13-00347-t002:** Parameter of the designed boost converter.

Variable	Parameter	Value
$U_{in}$	Input voltage	5–12 V
*D*	Duty cycle	0.514
*R*	Load resistance	50 Ω
$f_{sw}$	Switching frequency	25 MHz
*L*	Inductor	154 nH
$R_{L}$	Resistance of inductor at 25 MHz	0.84 Ω
$U_{out}$	Output voltage	9.6–22.8 V
$P_{out}$	Output power	1.9–10.7 W

**Table 3 micromachines-13-00347-t003:** Measurements of test samples at 25 ${}^{\circ }C$.

Measure	ISIT Sample	Inductor Ref1	Inductor Ref2
$V_{C}$ [$mm^{3}$]	2.04	4.87	0.48
$H_{max}$ [A/m]	$9.50\times 10^{5}$	$3.13\times 10^{5}$	$2.28\times 10^{5}$
$H_{sat}$ [A/m]	$6.60\times 10^{5}$	$2.05\times 10^{5}$	$1.04\times 10^{5}$
$B_{sat}$ [T]	2.02	0.65	0.42
$\mu_{max}$	6.1	7.0	8.4

**Table 4 micromachines-13-00347-t004:** Measurement results of the boost converter.

$U_{\mathrm{in}}$ [V]	$I_{\mathrm{in}}\left[A\right]$	$P_{\mathrm{in}}\left[W\right]$	$U_{\mathrm{out}}\left[V\right]$	$I_{\mathrm{out}}\left[A\right]$	$P_{\mathrm{out}}\left[W\right]$	η[%]	$T_{L}{[}^{\circ }C]$	$T_{T}{[}^{\circ }C]$
5.0	0.47	2.35	9.6	0.20	1.93	82.0	41	45
9.0	0.84	7.56	17.5	0.35	6.13	81.0	81	79
10.0	0.91	9.10	19.1	0.39	7.45	81.9	84	91
12.0	1.08	12.96	22.8	0.47	10.72	82.7	109	122

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
