# Peer review of "High Temperature Magnetic Cores Based on PowderMEMS Technique for Integrated Inductors with Active Cooling"

_micromachines, 2022, doi:10.3390/mi13030347_

Round 1
Reviewer 1 Report
This is an interesting study and the authors have collected a unique dataset using the cutting-edge methodology. The paper is generally well written and structured. Most importantly, authors achieved frequency efficiency up to 83% using a DC/DC converter with their fabricated micro inductor, which has an option to add Gan FET externally. In my opinion, the paper has no shortcomings in regards to some data analyses, modeling, and text, and I feel that this paper can be accepted as it is.
Author Response
Thanks for your review!
Reviewer 2 Report
this paper provide a novel MEMS-inductors with nanoparticle as the magnetic core, the device show a high performance under 400℃ with active cooling system. The research was planned well, and the experiment results are very helpful and informative for DC/DC convert area. This paper can be accepted after some contents are complemented.
- It could be better if add the parameter about the magnetic particle and coil dimension, such as the particle diameter, the coil length and the width, et al
- It could be better if this paper can give a deeply comment about the cooling of some important character of magnetic core such as fluid speed, different fluid medium et al
- Due to use the ALD Fabrication, the deposit rate of magnetic particle is low, is there any other way to fab the magnetic particles?
Author Response
Hi,
thanks for your review. My answers to your questions will follow:
1. It could be better if add the parameter about the magnetic particle and coil dimension, such as the particle diameter, the coil length and the width, et al
--> Not sure, if I understood this point correctly. Magnetic powder with particle size is mentioned in line 52f and in Figure 5, the sample dimensions are display. Which parameter should be added?
2. It could be better if this paper can give a deeply comment about the cooling of some important character of magnetic core such as fluid speed, different fluid medium et al.
--> Regarding the active cooling, the fluid speed is mentioned in line 208ff. But it's only a first experiment for this paper. Further investigations will follow, in best case, we will work with different speeds and gas and liquid fluids in the future.
3. Due to use the ALD Fabrication, the deposit rate of magnetic particle is low, is there any other way to fab the magnetic particles?
--> Of course there are other fabrication processes e.g. with polymer binder and so on but as mentioned in this paper our fabrication method based on ALD brings a lot of benefits in our eyes. Especially, the active cooling through the porous structure is only possible because this fabrication process.
Best regards,
Malte Paesler